# Performance Evaluation of the IR Biotyper^®^ System for Clinical Microbiology: Application for Detection of *Staphylococcus aureus* Sequence Type 8 Strains

**DOI:** 10.3390/antibiotics11070909

**Published:** 2022-07-07

**Authors:** Jun Sung Hong, Dokyun Kim, Seok Hoon Jeong

**Affiliations:** Department of Laboratory Medicine and Research Institute of Bacterial Resistance, College of Medicine, Yonsei University, Seoul 03722, Korea; jumphong2@nate.com (J.S.H.); kscpjsh@yuhs.ac (S.H.J.)

**Keywords:** *Staphylococcus aureus*, ST5, ST8, ST72, ST188, strain typing, MRSA, MSSA

## Abstract

Background: Methicillin-resistant *S. aureus* (MRSA) clonal lineages have been classified based on sequence type (ST) and pulsotype associated with human infection. Providing rapid and accurate epidemiological insight is important to address proper infection control in both community-acquired and nosocomial hospital settings. In this regard, this study was performed to evaluate the IR Biotyper^®^ (IRBT^®^) for strain typing of *S. aureus* clinical isolates on three media. Methods: A total of 24 *S. aureus* clinical isolates comprising 15 MRSA isolates (six ST5, three ST72, three ST8, and three ST188 isolates) and nine methicillin-susceptible *S. aureus* (MSSA) isolates (three ST5, three ST72, and three ST8 isolates) were included for evaluating the IRBT^®^. Molecular characterization of all *S. aureus* isolates was performed by conventional PCR and sequencing methods. The IRBT^®^ was evaluated according to manufacturer instructions and a modified sample procedure on commonly used BAP, MHA, and TSA media. Subsequently, the spectra obtained by IRBT^®^ software were compared with dendrograms of PFGE analysis. Results: In this study, the modified sample procedure for reducing the amount of bacteria and bacterial concentration improved the acquisition quality pass rate of the IRBT^®^. Each spectrum of *S. aureus* ST5, ST72, and ST188 isolates on all three media could not be clustered by IRBT^®^. However, the dendrogram obtained from the spectra of *S. aureus* ST8 isolates on TSA medium were in concordance with that obtained by PFGE analysis. In addition, the visual distribution of *S. aureus* ST8 isolates on TSA medium in a 2D scatter plot appeared as separated point set from those of *S. aureus* ST5, ST72, and ST188 isolates. Conclusions: The IRBT^®^ system is a rapid strain typing tool using the FTIR spectroscopic method. This system demonstrated the possibility of discriminating the strain types of *S. aureus* clinical isolates. Indeed, *S. aureus* ST8 isolates on TSA medium were successfully differentiated from other strain type isolates.

## 1. Introduction

*Staphylococcus aureus* has been commonly documented as one of the most important pathogens among gram-positive cocci in both community-acquired and hospital-acquired infections [1,2]. In particular, methicillin-resistant *S. aureus* (MRSA) is resistant to all β-lactam antimicrobials due to the expression of penicillin-binding protein 2a [3]. Furthermore, the Staphylococcal Chromosomal Cassette *mec* (SCC*mec*) element, coharboring resistance genes to different classes of antimicrobials, usually confers multidrug-resistant phenotypes in MRSA strains [4]. Therefore, the outbreak and spread of MRSA isolates have become a threat to public health worldwide [5,6,7].

MRSA clinical isolates are usually classified into several major clonal lineages by sequence type (ST) and pulsotype. Most ST5 MRSA isolates harboring SCC*mec* type II have been reported as a major clone associated with hospital-originated infection [8]. In contrast, most ST72 MRSA isolates carrying SCCmec type II have been found in community-acquired infections [9], and ST8 MRSA isolates carrying SCC*mec* type IV (USA300 pulsotype) are particularly reported to be an epidemic lineage of community-onset infection in the USA [8,10]. These clones, which are classified as highly virulent clones associated with human infection, are reported to be associated with poor clinical outcome in infected patients [11]. In addition to human-associated MRSA infections, MRSA ST188 isolates has been reported as one of the major lineages of livestock-associated MRSA infection [12]. Therefore, rapid epidemiological evaluation is important to address proper infection control and to prevent further dissemination in community and hospital settings.

Pulsed-field gel electrophoresis (PFGE) is one of the strain typing methods with high resolution [13]. In addition, multi-locus sequence typing (MLST) and the core genome MLST (cgMLST) approach enable sharing and comparison of loci among laboratories using the same schemes [14]. However, PFGE, MLST, and cgMLST nucleotide sequence-based strain typing techniques are time-consuming and require relatively high expertise, while this analysis can generate much more accurate genetic information.

In this regard, a phenotypic method based on the Fourier-transform infrared (FTIR) spectroscopy system has been introduced to detect dissemination of bacterial clones in gram-negative bacilli and gram-positive cocci [15]. The principle of FTIR-based typing is to determinate a variety of molecular vibration fingerprints such as lipids, nucleic acids, and proteins by quantifying the specific-absorption of infrared light [16]. This methodology allows easy, fast results within 4 h and a cost-effective workflow to the user for strain typing in a clinical microbiology laboratory.

In this study, we aimed to evaluate the performance for strain typing of *S. aureus* isolates using the IR Biotyper^®^ (IRBT, Bruker Daltonics GmbH & Co. KG, Bremen, Germany) launched in 2017 according to the type of culture medium and sample preparation and to compare the results for clustering *S. aureus* isolates with those by PFGE analysis.

## 2. Materials and Methods

### 2.1. Study Design

For this study, 24 *S. aureus* clinical isolates were used to evaluate the performance of the Bruker IRBT^®^. For evaluating the IRBT^®^ on a variety of culture media, all 24 *S. aureus* isolates were grown on three types of commonly used blood agar plate (BAP; BANDIO Co., Ltd., Pocheon-si, Gyeonggi-do, Korea), tryptic soy agar (TSA; BANDIO Co., Ltd., Pocheon-si, Gyeonggi-do, Korea), and Mueller-Hinton Agar (MHA; BANDIO Co., Ltd., Pocheon-si, Gyeonggi-do, Korea) for 18 h at 37 °C. Dendrograms built with spectra generated by the IRBT^®^ were compared with STs and pulsotypes of MLST and PFGE analyses.

### 2.2. Bacterial Profiles

A total of 24 *S. aureus* clinical isolates including 15 MRSA isolates (six ST5, three ST72, three ST8, and three ST188 isolates) and nine MSSA isolates (three ST5, three ST72 and three ST8 isolates) were used for evaluating the IRBT^®^ in this study. Detailed molecular characterizations of the strains are summarized in Table 1. Before the experiments, bacterial identification was confirmed by matrix-assisted laser desorption/ionization time-of-flight mass spectrometry (MALDI Biotyper; Bruker Daltonics GmbH & Co. KG, Bremen, Germany) at the species level. The antimicrobial susceptibility (AST) of all 24 *S. aureus* isolates was confirmed by a disk diffusion test using a 30 μg cefoxitin disk (Oxoid Ltd., Basingstoke, UK), in accordance with the Clinical and Laboratory Standards Institute guidelines [17]. The molecular characterization of isolates including *mec*A, *mec*C, SCC*mec* type, *spa* type, TSST-1 (*tst-1*), and PVL (*pvl*) was assessed by conventional PCR and sequencing methods (Appendix A). For strain typing, MLST experiments were performed by comparing the partial sequences of the seven housekeeping genes *arcC*, *aroE*, *glpF*, *gmk*, *pta*, *tpi*, and *yqiL* gene, to sequences in the *S. aureus* MLST database (https://pubmlst.org/saureus/, accessed on 6 June 2022) to determine allelic types and sequence types (STs) of the *S. aureus* isolates.

### 2.3. IR Biotyper^®^ Protocol

All MRSA and MSSA isolates were grown at 37 °C on BAP, TSA, and MHA for 24 ± 0.5 h. First, a loopful of bacterial culture (1 μL) was collected and suspended in 50 μL of 70% ethanol in a 1.5 mL e-tube containing sterile metal rods. To obtain a uniform suspension, a multi-vortexer was used. After vortexing for 1 min, 50 μL of sterile water was added, and the final volume of 100 μL of solution was vortexed for 1 min. Second, 15 μL of the bacterial suspension was spotted onto the IRBT silicon plate and dried at 37 °C together with 12 μL of two spots of each IR test standard 1 (IRTS1) and IR test standard 2 (IRTS2) suspension, which were contained in the IRBT^®^ kit as quality controls, until a dry film formed over 30 min. Then, 15 μL of the suspension was used for each of three replicates for technical triplicates of each sample. Lastly, the dried silicon plate was inserted into the IRBT spectrometer (Bruker Daltonics GmbH & Co. KG, Bremen, Germany) at default analysis settings. Spectra of isolates were acquired using OPUS 7.5 software (Bruker Daltonics GmbH & Co. KG, Bremen, Germany). The spectra that met the default quality criteria of absorption [0.4 arbitrary unit (AU) < *D* value < 2 AU], signal/noise (<150 × 10^−6^ AU), signal/water (<300 × 10^−6^ AU), and fringes (<100 × 10^−6^ AU) were determined as “quality pass” in the IRBT^®^ analysis. The *S. aureus* isolates that failed to pass the default quality criteria were re-examined. If the spectra did not meet the default quality criteria after re-examination, those with failure only for absorption criteria were included in the analysis. The spectra obtained with “quality pass” were analyzed for building the dendrograms. The software contains a feature that automatically proposes a cut-off value that defines up to which distance spectra are considered to be in the same cluster.

### 2.4. Pulsed-Field Gel Electrophoresis (PFGE)

The *Sma*I macrorestriction banding patterns of *S. aureus* isolates were evaluated by PFGE analysis as previously described [3]. A single colony of *S. aureus* isolate was inoculated into 2 mL of brain heart infusion broth and incubated with vigorous shaking at 37 °C for 24 h. Five hundred microliters of the adjusted cell suspension were centrifuged at 12,000× *g* for 5 min, and the supernatant was aspirated. The pellet was suspended in 300 μL of TE buffer and equilibrated at 37 °C for 10 min. Four microliters of lysostaphin and 300 μL of SeaKem Gold agarose (FMC, Rockland, ME) in TE buffer (equilibrated to 55 °C) were added to the cell suspension and dispensed into the wells of a plug mold. The plugs were allowed to solidify in the refrigerator (4 °C) for 10 min, placed into a tube containing at least 1 mL of lysis buffer, and incubated at 37 °C for at least 4 h. The lysis buffer was decanted, and 1 mL of TE buffer was added. The TE buffer washing was repeated at least three more times, and the plugs were stored at 4 °C. A plug slice was equilibrated in 1× restriction buffer for at least 40 min. After removal of the 1× restriction buffer, 1 μL of *Sma*I restriction enzyme (Takara Co., Ltd., Tokyo, Japan) in 100 μL of 1× restriction buffer was added to each tube, and the tubes were incubated at 25 °C for 4 h. PFGE was performed using a CHEF-DRII system (Bio-Rad, Hercules, California, USA) using the following run parameters: 200 V (6 V/cm); temperature, 14 °C; initial switch, 5.3 s; final switch, 34.9 s; and time, 20 h. *S. aureus* ATCC BAA1681 (USA100 pulsotype) and *S. aureus* ATCC BAA1768 (USA800 pulsotype) were used as reference strains.

## 3. Results

### 3.1. Optimization of IRBT^®^ Sample Preparation

When we performed IRBT^®^ analysis following the manufacturer’s instructions, the acquisition quality pass rate of *S. aureus* isolates collected from BAP, MHA, and TSA media was 51.4% (37/72), 25.0% (18/72), and 22.2%, (16/72), respectively. Absorption values of most of the spots exceeded the default quality criteria (0.4 AU < value < 2 AU). Thus, the cut-off value calculated by the OPUS 7.5 software could not cluster the 24 *S. aureus* isolates belonging to four STs in a dendrogram, which indicated inability to divide clusters by IR type.

To improve the acquisition quality pass rate, we modified the procedures as follows. First, we spotted 12 μL instead of 15 μL of bacterial suspension from all three agar media onto silicon plates to reduce the amount of bacterial suspension. Second, the sample preparation sequence was changed from an EtOH-water method to a water-EtOH method [16,18]. Lastly, the amount of bacteria was reduced from a 1 μL loopful of bacterial culture to 0.5 μL loopful of bacterial culture. After these modifications, the acquisition quality pass rate was successfully increased to 68.1% (49/72) on BAP medium, 98.6% (71/72) on MHA medium, and 100% (72/72) on TSA medium.

### 3.2. Comparison of the Results by IRBT^®^ According to Media

After obtaining the spectra acquisition, dendrograms of 24 *S. aureus* isolate spectra were constructed by IRBT^®^ spectroscopy analysis and are presented in Figure 1. Based on analysis of three technical replicates in this study, the IRBT^®^ created 24 average spectra from 72 spectra of 24 *S. aureus* isolates on each of the three media. The cut-off value automatically calculated by the OPUS 7.5 software was 0.128 on BAP medium, 0.111 on MHA medium, and 0.119 on TSA medium (Figure 1). The cut-off value was set at 0.215 for the same conditions on a variety of culture media. The spectra on BAP medium were clustered into four IR types (Figure 1A). The IRBT^®^ on MHA medium can differentiate 24 *S. aureus* isolates into three IR types (Figure 1B). On TSA medium recommended by the manufacturer, the spectra were clustered into three IR types (Figure 1C). When we combined the above 72 average spectra on three media at cut-off value of 0.215, the dendrograms were divided into 10 IR types (Figure 2).

### 3.3. Strain Typing of IRBT^®^ Compared with the Results of PFGE Analysis

PFGE analysis was performed to evaluate the IRBT^®^ (Figure 3). As a reference method for strain typing by MLST and PFGE analyses, 24 *S. aureus* isolates of four STs were divided into four PFGE banding patterns based on a similarity cut-off of about 80% (pulsotypes A–D) as follows: (i) Pulsotype A included three MSSA ST5/spa-t688 isolates, (ii) Pulsotype B included six MRSA ST5/spa-t002 and -t2460 isolates with two reference strains (USA100 and USA800 pulsotype clone), (iii) Pulsotype C included three MRSA ST188/spa-t189 isolates, and (iv) Pulsotype D1 included three MRSA and three MSSA ST8/spa-t008 isolates, and Pulsotype D2 included three MRSA ST72/spa-t324 and three MSSA ST72/spa-t126 isolates.

The IRBT^®^ on three media could not classify *S. aureus* ST5, ST72, and ST188 isolates regardless of methicillin-resistance phenotype because the spectra were divided into only one or two IR types compared to their pulsotypes by PFGE analysis (Figure 3). Only the IR type of the spectra of six *S. aureus* ST8/spa-t008 isolates on TSA media showed concordance with pulsotype D1 of PFGE analysis, but one isolate from BAP medium (SA12 strain) and one isolate from MHA medium (SA 15 strain) were not clustered into same IR type with other ST8 isolates. An assessment of 2D scatter plot for entire spectra of linear discriminant analysis (LDA) on TSA medium, which showed the best results in differentiating *S. aureus* ST8/spa-t008 isolates, presented the variability and similarity among the 72 spectra of 24 *S. aureus* isolates, as in Figure 4. The visual distribution of 24 *S. aureus* isolates belonging to four pulsotypes of PFGE analysis appeared as four separate point sets. The MRSA CC5 and *S. aureus* CC8 isolates were the furthest from the other isolates including MSSA ST5, *S. aureus* ST72, and *S. aureus* ST188.

An additional experiment using 20 MRSA isolates including eight ST8, three ST1, three ST89, three ST403, and three ST632 isolates was performed to evaluate the reproducibility of clustering of *S. aureus* ST8 isolates by IRBT^®^ in two independent runs on separate days (Appendix A), and eight ST8 MRSA isolates were clustered from other STs in the dendrogram with the same cut-off value of 0.215, and from a 2D scatter plot (Appendix A).

## 4. Discussion

*S. aureus* has become a common threatening pathogen for hospital and community-acquired disease associated with human and livestock infections [19,20]. MRSA has been increasingly disseminated during the last 20 years in South Korea [21,22]. In particular, MRSA ST5 and ST72 isolates have been classified as major virulent lineages [3,23]. In addition to ST5 and ST72 clones, ST8 PVL-positive MRSA isolate (USA300 clone), which has been initially observed in the USA, has been newly observed since the early 2010s in South Korea [3,10,21,23,24,25]. This clone was increasingly reported in South Korea [21], suggesting a potential concern for spread of a putative virulent factor in hospital and community settings. Therefore, rapid detection and strain typing are important for proper infection control and for preventing further dissemination. In this regard, FTIR spectroscopy techniques could be an alternative for rapid and accurate strain typing. To date, *Klebsiella pneumoniae*, *Acinetobacter baumannii*, and *Lactiplantibacillus plantarum* have been evaluated for strain typing using the IRBT^®^ [16,18,26]. However, this method is challenging for *S. aureus* strain typing. In this study, optimization of the IRBT procedure including bacterial amount, sample preparation, and appropriate media choice for *S. aureus* strains was performed, and the accuracy of *S. aureus* clinical isolate strain typing using the IRBT^®^ was evaluated by comparison with the results of PFGE analysis, which is the gold-standard strain typing method.

The IRBT^®^ spectra showed poor acquisition quality pass rates (51.4% on BAP medium, 25.0% on MHA medium, and 22.2% on TSA medium) when following the manufacturer’s instructions. In this study, we prepared samples through a modified procedure to reduce the amount of bacteria (from 1 μL to 0.5 μL loopful of bacterial culture) and bacterial concentration (from 15 μL to 12 μL spotting) because most *S. aureus* isolates showed failed sample quality due to high absorption. With these modifications, the acquisition quality pass rate was significantly improved on all three media. Subsequently, spectra were successfully obtained and dendrograms were generated to classify IR types (Figure 1). These results of modified sample preparation suggest that controlling the amount of bacteria and bacterial concentration will provide better solutions when using the IRBT^®^ with *S. aureus* isolates.

FTIR spectroscopy systems are known to be affected by features of bacterial colonies, which are affected by the culture media [18]. TSA medium was recommended for *S. aureus* isolates according to the manufacturer’s instructions. To evaluate the quality of spectra according to the type of culture media, we examined the IRBT^®^ with three commonly used culture media (BAP, MHA, and TSA) in laboratory microbiological medicine. The acquisition pass rate was highest in isolates on TSA, and the six spectra of *S. aureus* ST8 isolates on TSA medium were clustered into the same clade, which is consistent with the pulsotypes of those isolates (Figure 1 and Figure 3). In contrast, the spectra of *S. aureus* ST8 isolates on BAP and MHA medium were not clustered into one clade by IRBT^®^. In addition, the spectra of each isolate from the three culture media were not clustered into one clade (Figure 2) but were clustered by type of medium. This finding suggests that the results of IRBT^®^ were influenced by the features of bacterial colonies due to the composition of the media, and that TSA medium could be a reference media for *S. aureus* typing by IRBT^®^. A similar observation was reported by Złoch et al. that the molecule profiles of *S. aureus* isolates showed different protein spectra on variations of culture medium using MALDI-TOF MS analysis [27].

The IRBT^®^ could not differentiate phenotypic and genotypic variation related with strain type, methicillin-resistance, and *S. aureus* toxin due to the overlapping point sets in 2D scatter plots (Figure 4). However, *S. aureus* ST8 isolates on TSA medium were separately clustered from ST5, ST72, and ST188 isolates by the IRBT^®^ system. In a previous study, the clustering of strain type using IRBT^®^ showed high discrimination power for *K. pneumoniae* ST11 isolates, and clinical isolates of *A. baumannii* from the nosocomial outbreak showed high concordance with the results of PFGE analysis [16,26]. Even though low discriminating power for ST5 and ST72 *S. aureus* isolates was identified, the IRBT^®^ is expected to help classify a specific clone such as ST8 of *S. aureus* in laboratory microbiological medicine.

Most of ST8 MRSA isolates in South Korea were PVL-positive t008 SCCmec type IV strains (USA300 clone), while PVL-negative ST8-MRSA-SCCmec type IV, Lyon Clone/UK-EMERSA-2 lineage is prevalent in Europe, mainly isolated in hospital-associated infections [24]. Therefore, the combination of an ST8 profile by IRBT^®^ system and an absence of PVL by PCR could indicate either a Lyon Clone/UK-EMRSA-2 or an excision of the PVL phage from a USA300 strain.

A limitation in this study is the lack of diversity of tested *S. aureus* isolates. The number of isolates in one run was limited due to three technical replicates in one silicon plate. However, twenty-four *S. aureus* isolates with four major STs are included as representatives of human- and livestock-association infection, and additional four minor STs were included. Furthermore, the MRSA clinical isolates are not diverse, and three MRSA lineages of ST5, ST8, and ST72 tested in this study are globally the most common clones causing human-associated infections.

The IRBT^®^ system is a rapid strain typing tool using the FTIR spectroscopic method, and this system demonstrated the possibility of discriminating the strain types of *S. aureus* clinical isolates. Indeed, spectra from ST8 *S. aureus* isolates on TSA medium were separately clustered from ST5, ST72, and ST188 isolates by the IRBT^®^ system. Further evaluation should be performed to clarify the usefulness of this system.

## Figures and Tables

**Figure 1 antibiotics-11-00909-f001:**
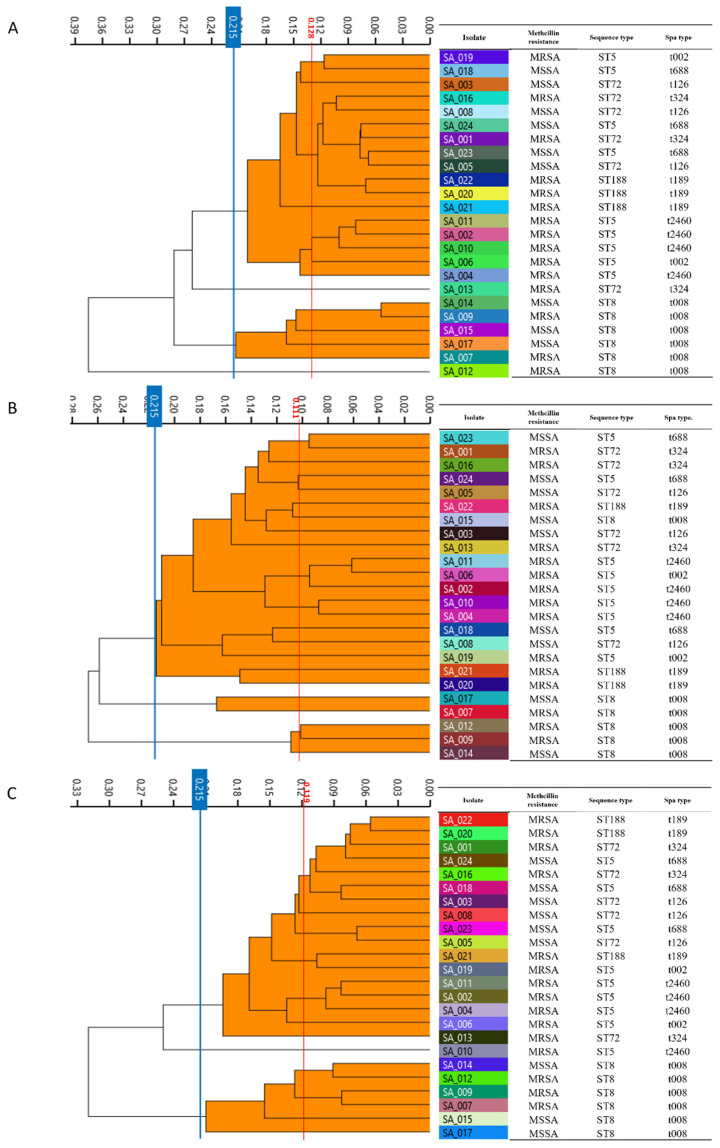
Dendrograms of 24 *S. aureus* isolates based on the IRBT^®^ spectra on three media at cut-off of 0.215 with bacterial molecular profiles. The cut-off value automatically generated by the OPUS 7.5 software was indicated with red vertical line. (**A**) 24 spectra of *S. aureus* isolates on BAP medium by averaging three technical replicates, (**B**) 24 spectra of *S. aureus* isolates on MHA medium by averaging three technical replicates, (**C**) 24 spectra of *S. aureus* isolates on TSA medium by averaging three technical replicates.

**Figure 2 antibiotics-11-00909-f002:**
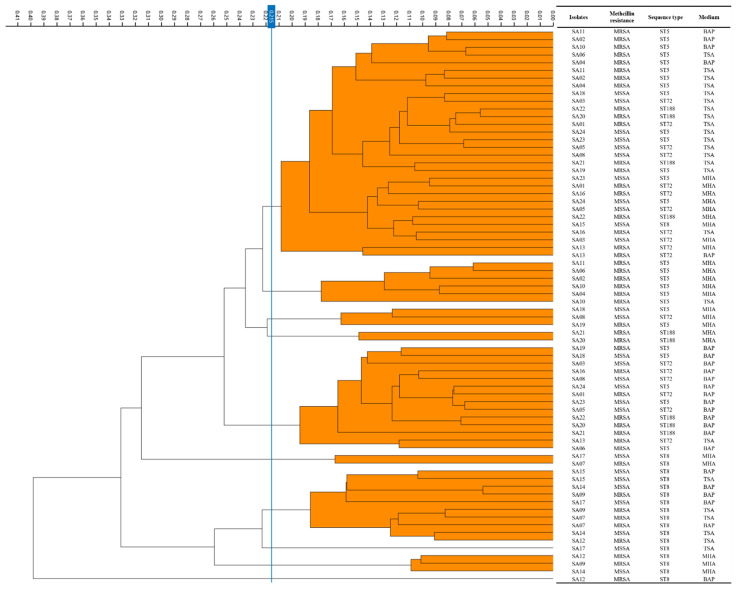
Dendrogram of 72 average spectra of 24 *S. aureus* isolates from three media at cut-off of 0.215 comparing with bacterial molecular profiles.

**Figure 3 antibiotics-11-00909-f003:**
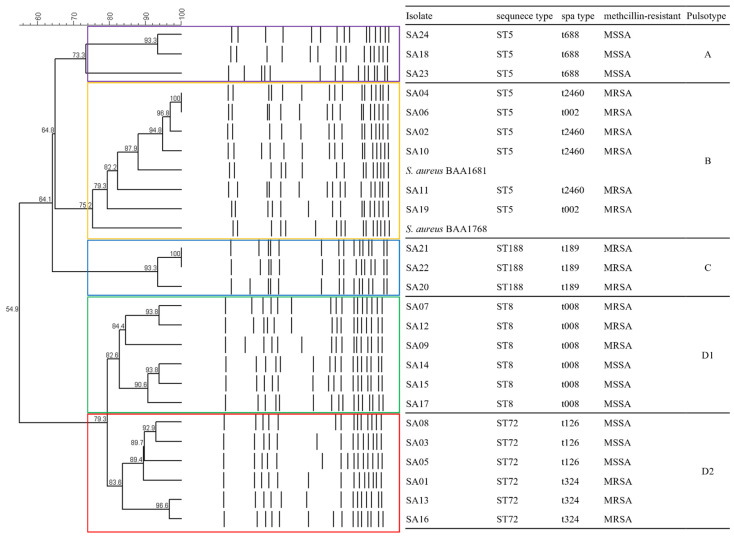
Characteristic of 24 *S. aureus* isolates with dendrogram of PFGE analyses. Purple, yellow, blue, green, and red box mean each clustering pulsotypes.

**Figure 4 antibiotics-11-00909-f004:**
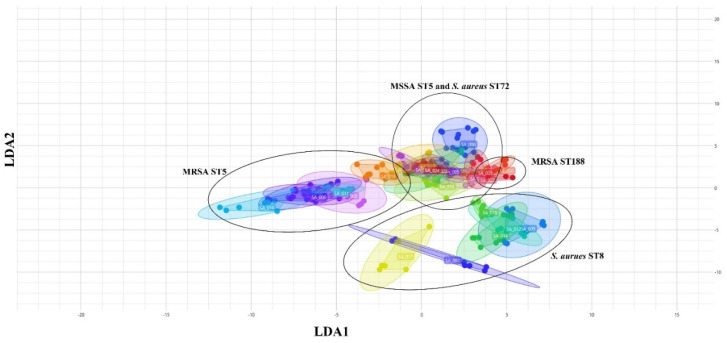
Assessment of 2D scatter plot for 72 spectra of LDA analyses on TSA medium. MRSA ST5 and *S. aureus* ST8 are separated from MSSA ST5, MRSA ST188, and *S. aureus* ST72.

**Table 1 antibiotics-11-00909-t001:** Molecular data for the isolates tested in this study.

Strain	Strain Type	Methicillin-Resistance	*mecA*	*mecC*	SCC*mec*	TSST-1	PVL	spa Type
SA01	ST72	R	P	N	IV	N	N	t324
SA02	ST5	R	P	N	II	P	N	t2460
SA03	ST72	S	N	N	-	N	N	t126
SA04	ST5	R	P	N	II	P	N	t2460
SA05	ST72	S	N	N	-	N	N	t126
SA06	ST5	R	P	N	II	P	N	t002
SA07	ST8	R	P	N	IV	N	P	t008
SA08	ST72	S	N	N	N	N	N	t126
SA09	ST8	R	P	N	IV	N	P	t008
SA10	ST5	R	P	N	II	P	N	t2460
SA11	ST5	R	P	N	II	N	N	t2460
SA12	ST8	R	P	N	IV	N	P	t008
SA13	ST72	R	P	N	IV	N	N	t324
SA14	ST8	S	N	N	-	N	N	t008
SA15	ST8	S	N	N	-	N	N	t008
SA16	ST72	R	P	N	IV	P	N	t324
SA17	ST8	S	N	N	-	N	N	t008
SA18	ST5	S	N	N	-	N	N	t688
SA19	ST5	R	P	N	II	N	N	t002
SA20	ST188	R	P	N	V	N	N	t189
SA21	ST188	R	P	N	IV	N	N	t189
SA22	ST188	R	P	N	V	N	N	t189
SA23	ST5	S	N	N	-	N	N	t688
SA24	ST5	S	N	N	-	N	N	t688

Abbreviations: SCC*mec*, Staphylococcal chromosomal cassette *mec*; TSST-1, toxic shock syndrome toxin-1; PVL, Panton-Valentine leucocidin; ST, sequence type; R, resistance; S, susceptibility; P, positive; N, negative.

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
