# Peer review of "Performance Evaluation of the IR Biotyper® System for Clinical Microbiology: Application for Detection of Staphylococcus aureus Sequence Type 8 Strains"

_antibiotics, 2022, doi:10.3390/antibiotics11070909_

Round 1
Reviewer 1 Report
Unfortunately, the research results were not significant and do not worth being published in the journal "Antibiotics".
I would like to highlight that the article is well written and readable. Unfortunately, the conclusions of the article (ST8 differentiation) come to me at least very preliminary and optimistic based on the presented results. The authors themselves admit that the results obtained were not reproducible (lines 270-273: …..lack of the reproducibility among different runs).
In addition to modifying the protocol so that it is reproducible, I would definitely recommend verifying statement that ST8 sequence type of S. aureus on a higher number of isolates of different MLST profiles (mainly ST8).
Furthermore, the assertion that MRSA ST8 isolates are normally Panton-Valentine leucocidin (PVL) positive ([10]; lines 221-222) contradicts selected isolates analyzed in this work (Table 1).
The molecular characterization of isolates including mecA, mecC, SCCmec type, spa type, TSST-1 (tst-1), and PVL (pvl) was assessed by conventional PCR and sequencing methods. It is not described
Author Response
I would like to highlight that the article is well written and readable. Unfortunately, the conclusions of the article (ST8 differentiation) come to me at least very preliminary and optimistic based on the presented results. The authors themselves admit that the results obtained were not reproducible (lines 270-273: …..lack of the reproducibility among different runs).
Answer: Thank you for your valuable comment. The intended meaning of “Lack of the reproducibility among different runs” were that the results obtained by different runs could not be combined and analyzed. To evaluate the reproducibility of ST8 differentiation, additional experiments were performed as separate runs on two independent days, and results are shown in Line 212-216 and Figure S1.
In addition to modifying the protocol so that it is reproducible, I would definitely recommend verifying statement that ST8 sequence type of S. aureus on a higher number of isolates of different MLST profiles (mainly ST8).
Answer: Following the comment, additional experiment using 20 MRSA isolates including eight ST8, three ST1, three ST89, three ST403, and three ST632 isolates was performed, and ST8 MRSA isolates were clustered from other STs in the dendrogram with the same cut-off value of 0.215, and a 2D scatter plot (Figure S1C).
Furthermore, the assertion that MRSA ST8 isolates are normally Panton-Valentine leucocidin (PVL) positive ([10]; lines 221-222) contradicts selected isolates analyzed in this work (Table 1).
Answer: All MRSA ST8 isolates included in this study were PVL positive isolates, and three MSSA ST8 isolates were PVL negative. In addition, eight more ST8 MRSA isolates exhibiting PVL positive were additionally tested (Figure S1).
The molecular characterization of isolates including mecA, mecC, SCCmec type, spa type, TSST-1 (tst-1), and PVL (pvl) was assessed by conventional PCR and sequencing methods. It is not described
Answer: Following the comment, the detailed information of primers used in molecular characterization are added as Table S1.
Reviewer 2 Report
The paper “Performance evaluation of the IR Biotype system for clinical microbiology: Application for detection of Staphylococcus aureus sequence type 8 strains” is interesting and well written.
I have just some minor concerns.
Line 15. The acronym MSSA has not been previously reported in full. The abstract should be self supporting.
Line 64: I suppose that the authors mean “principle” instead of “principal”.
Line 90: which was the cefoxitin disk concentration?
Line 153-160: on which bases were the procedure modifications chosen?
Author Response
The paper “Performance evaluation of the IR Biotype system for clinical microbiology: Application for detection of Staphylococcus aureus sequence type 8 strains” is interesting and well written.
Line 15. The acronym MSSA has not been previously reported in full. The abstract should be self supporting.
Line 64: I suppose that the authors mean “principle” instead of “principal”.
Answer: We apologize the error. The manuscript was revised following the comments in line 16 and 67.
Line 90: which was the cefoxitin disk concentration?
Answer: Thank you for your comment. We added the concentration of cefoxitin disk (30 μg) in line 93.
Line 153-160: on which bases were the procedure modifications chosen
Answer: We modified the sample procedure in three points: 1) reducing the amount of bacteria from 1 μL loopful to 0.5 μL loopful, 2) reducing the spotting volume of bacterial suspension from 15 μL to 12 μL and 2) sample preparation sequence from EtOH-distilled water to distilled water-EtOH. The first and second modifications were based on our initial experiment exceeding the quality cut-off of absorption value which indicate the amount or concentration of bacterial suspension was high. The third modification was based on the previous studies (Reference 16 and 18).
Reviewer 3 Report
In this study, optimization of the IRBT® procedure including bacterial amount, sample preparation, and appropriate media choice for S. aureus strains was performed, and the accuracy of S. aureus clinical isolate strain typing using the IRBT® was evaluated by comparison with the results of PFGE analysis. Based on this assumption, specific comments related to the article are described below.
In abstract: please rephase the abstract.
Line 75-76: The number of strains is too small and the reasons for selection
Line 91-97: the primers of these genes (mecA, mecC…)?
Line110: The trade mark “®” was omitted. Please make sure that the names of “IRBT®” are consistent.
Line 188: the cut-off value of PFGE banding patterns.
Author Response
In this study, optimization of the IRBT® procedure including bacterial amount, sample preparation, and appropriate media choice for S. aureus strains was performed, and the accuracy of S. aureus clinical isolate strain typing using the IRBT® was evaluated by comparison with the results of PFGE analysis. Based on this assumption, specific comments related to the article are described below.
In abstract: please rephase the abstract.
Answer: Following the comment, the abstract was revised to clarify the meaning.
Line 75-76: The number of strains is too small and the reasons for selection
Answer: Thank you for your valuable comment. The number of isolates in one run were limited due to three technical replicates in one silicon plate. Twenty-four S. aureus isolates with four major STs are included as representatives of human- (ST5, ST8, ST72) and livestock-association infection (ST188). MRSA lineages of ST5, ST8, and ST72 are globally the most common clones causing human-associated infections. Furthermore, additional experiment using 20 MRSA isolates including eight ST8, three ST1, three ST89, three ST403, and three ST632 isolates was performed.
Line 91-97: the primers of these genes (mecA, mecC…)?
Answer: Following the comment, the detailed information of primers used in molecular characterization are added as Table S1.
Line110: The trade mark “®” was omitted. Please make sure that the names of “IRBT®” are consistent.
Answer: We apologize the error. The manuscript was revised following the comments in line 113.
Line 188: the cut-off value of PFGE banding patterns.
Answer: Following the comment, the cut-off value of PFGE banding patterns were added in line 191.
Round 2
Reviewer 1 Report
No comments
Author Response
Thank you for reviewing our manuscript.
Reviewer 3 Report
No comment.
Author Response
Thank you for reviewing our manuscript.